# Impact of face masks and sunglasses on emotion recognition in South Koreans

**Garam Kim**[1], **So Hyun Seong**[1], **Seok-Sung Hong**[2], **Eunsoo Choi**[1] *

1 School of Psychology, Korea University, Sungbuk-gu, Seoul, South Korea, 2 Department of IT Psychology, Ajou University, Yeongtong-gu, Suwon, South Korea

* taysoo@korea.ac.kr

## Abstract

Due to the prolonged COVID-19 pandemic, wearing masks has become essential for social interaction, disturbing emotion recognition in daily life. In the present study, a total of 39 Korean participants (female = 20, mean age = 24.2 years) inferred seven emotions (happiness, surprise, fear, sadness, disgust, anger, surprise, and neutral) from uncovered, mask-covered, sunglasses-covered faces. The recognition rates were the lowest under mask conditions, followed by the sunglasses and uncovered conditions. In identifying emotions, different emotion types were associated with different areas of the face. Specifically, the mouth was the most critical area for happiness, surprise, sadness, disgust, and anger recognition, but fear was most recognized from the eyes. By simultaneously comparing faces with different parts covered, we were able to more accurately examine the impact of different facial areas on emotion recognition. We discuss the potential cultural differences and the ways in which individuals can cope with communication in which facial expressions are paramount.

## Introduction

Without doubt, we all have had trouble identifying emotional expressions from others in recent years when mask wearing became the norm. In this COVID-19-era, we now live in an environment where wearing masks is a necessity in our daily lives. As a result, we often encounter faces that are only partially exposed, impairing our daily social interactions due to our diminished ability to recognize facial expressions and their associated emotions. Thus, it is particularly important to understand the specific ways in which the ability to correctly infer emotions is restricted when parts of the face are occluded in this unusual time.

Overall, researchers agree that facial expression recognition is hindered when parts of the face are covered. The two key areas of the face that are important for reading facial expressions are the mouth and eyes [1–4]. There have been debates about which is more important, either the eyes or the mouth, in recognizing facial expressions. Notably, emotion type is considered to play a critical role in determining the key facial areas for reading emotions.

Past research focusing on six basic emotions (i.e., happiness, sadness, fear, anger, disgust, and surprise) has found that some emotions render consistent results, while others do not. First, studies on happiness or fear reported consistent results such that the mouth plays a key role in recognizing happiness [5–8] and the eyes in fear recognition [9–13]. Second, as for

**Funding:** E.C received the grant from School of Psychology, Korea University (K2110491). The funders had no role in study design, data collection and analysis, decision to publish, or preparation of the manuscript.

**Competing interests:** The authors have declared that no competing interests exist.

disgust, a growing number of studies have shown that the mouth has a greater effect than the eyes [11, 12]. Other basic emotions such as surprise, sadness, and anger showed mixed results [6–8, 11–14]. For instance, in detecting sadness, some studies showed that covering the mouth affects facial expression recognition [8], while others have shown more areas including the mouth, eyebrows, eyes, and rigid head posture also contribute to this [5]. Given the current state of the literature, further research is needed.

Some recent studies that empirically confirmed the impact of facial masks, thereby increasing the realism during the COVID-19 outbreak, were noticeable [9, 15–17]. As reviewed by Pavlova & Sokolov [18], recent studies that examined the effects of mask-wearing showed that mask hurts emotion recognition and that the impairment in recognition varied by specific emotions. Carbon [9] conducted the first study to test the effects of mask on recognition for six basic emotions (anger, disgust, fear, happiness, neutral, and sadness) with German adults. He found that when people recognize angry, disgusted, happy, and sad facial expressions, their performance was more impaired under the mask condition than under the no-mask condition, suggesting that the mouth area provides an important source of information for inferring these emotions from the face. However, there was no significant difference between the conditions in recognition for fear, indicating that the eyes (with and without a visible mouth) were sufficient for detecting fear. This study sheds light on the effects of wearing a mask, that is, the effects of occlusion of the mouth area on the recognition of different emotions in everyday contexts. These findings were also replicated with children age 9–10 years [15].

One limitation of these studies, however, is that the effects of eyes, which are critical in emotional expression and recognition of faces, were not examined. These studies make it difficult to identify whether the impairment is entirely due to the lack of information from the mouth or additional interference from the eyes. Alternatively, disturbance in emotion recognition for faces covered with masks may be buffered by the information obtained from the eyes. Thus, it would be important to make direct and simultaneous comparisons between the mouth-covered, eye-covered, and uncovered conditions, which would allow us to determine which facial area is important, and whether it depends on types of emotions.

Recently, there have been few studies that addressed this limitation and examined the effects of occlusion of both the mouth and the eyes on emotion recognition [16, 17]. Most notably, Noyes et al. [17] conducted an experiment that included the mask, sunglasses, and control conditions with UK adults. They found that the overall emotion recognition accuracy was the lowest when the mouth was occluded. However, the effects of the conditions differed by specific emotions. For instance, the effects of mask condition and sunglasses condition on the recognition of angry faces were not significantly different, and the mask condition rendered a greater accuracy rate than the sunglasses condition for sad faces. However, these findings seem inconsistent. For example, Carbon's study [9] showed that there was an impairment in recognition for sadness under masked conditions. Thus, more research is required to test whether the findings are robust.

Finally, one glaring limitation of prior research is that the participants were primarily from Western cultural contexts. It has generally been found that there is a cultural difference in emotional recognition between Asians and Westerners [19–21]. Specifically, previous studies document that East Asians tend to focus on the eyes rather than the mouth, whereas Westerners tend to focus on the entire face [22]. Other studies have shown that East Asians concentrate more on the eye area (upper face) compared to Westerners [10, 21, 23]. If this were the case, then Asians' emotion recognition for mask wearing faces would be less impaired, but somewhat more impaired for faces covered with sunglasses. Moreover, whereas Westerners are more accustomed to faces with sunglasses than to masked faces [17], East Asians have been wearing masks since pre-COVID-19 and are relatively more familiar with masked faces [24].

Also, the effects of masks or sunglasses on reading other person's facial expressions may differ depending on the cultural context (for a review see [18]). As an example, consider the findings that a face covered with Islamic headdresses such as niqāb impacts the recognition of emotions differently by cultural groups [13, 25], suggesting that a culturally attached meaning of head-dress may play a role. As for East Asians, they are not only less accustomed to sunglasses than Westerners, but sunglasses are often considered rude in interpersonal relationships [26, 27]. Given such cultural background, thus, it is necessary to test whether the effects of masks and sunglasses on facial expression recognition that are documented with Western participants would also apply to East Asians.

## Present study

Building on past studies, the present study aims to fill the gaps in previous research. To do this, we first conducted an integrative investigation that examined the effects of both the mouth and eye regions with realistic facial occlusions (i.e., mask and sunglasses) on six basic emotions. Second, going beyond the past studies that have been primarily conducted in the West, we aimed to test whether similar findings are observed in non-Western samples. This is important because many previous studies have demonstrated that there are cultural differences in the areas of the face to which people pay attention.

In the present study, we used a within-subject design with Korean undergraduate students as participants. To enhance the realism of the stimuli, we took steps to edit real-world masks and sunglasses images to face stimuli rather than simply cutting or covering images with a black box, as in previous studies [8, 13, 14, 28]. In addition to the main research question, we tested the effect of sex on facial expression recognition. According to previous studies, women appear to perform better than men when distinguishing facial expressions in faces covered with masks [29, 30]. In addition, it was found that individuals better identified (and labeled) emotions from facial expressions when the target was of different sex [31]. Thus, we examined whether emotion recognition differs according to the sex of the participant and the sex of the face stimuli.

## Methods

### Participants

Based on a power analysis (G-power software version 3.1), a total of 28 participants were required for the current experiment, which was designed with a repeated-measures ANOVA. Given an expected effect size of .25, and $\alpha = 0.05$, this led to an acceptable power of .8 [32]. However, we sampled more participants than the required number for potential participants who would get excluded if computers malfunction or those who did not meet the pre-set exclusion criteria (less than a 50% correct facial expression recognition answer rate when presented with a fully visible face without a face mask or sunglasses were excluded). This pre-determined criterion was based on that of Carbon's method [9]. Through an advertisement post on a Korean university community website, 40 students voluntarily registered for the study. All participants had normal or corrected-to-normal vision with no abnormal color discrimination ability. Participants received a 10,000 won (approximately 9 dollars) beverage gift card for participation in the study. This study was approved by the Institutional Review Board of Korea University (KUIRB-2020-0317-01). During the analysis, participants with less than a 50% correct facial expression recognition answer rate when presented with a fully visible face without a face mask or sunglasses were excluded. With 39 participants exceeding a 50% correct answer rate (average = 0.67), the final sample consisted of 39 participants (male = 19,

female = 20, $M_{age}$ = 24.2 years, $SD_{age}$ = 4.7 years). The data can be accessed at https://osf.io/fcg4d/.

## Materials

The Korean facial expression data used in the study was obtained from the Korea University Facial Expression Collection (KUFEC) [33]. The KUFEC was developed to minimize the cross-race effect by using the faces of Korean models [34]. Through an agreement between two researchers, six male and female models with relatively more accurate facial expressions were selected from a total of 49 models. Each identity posed a facial expression depicting happiness, surprise, fear, sadness, disgust, anger, and a neutral emotion. These were then photoshopped using Adobe Photoshop CC 2020 by adding surgical face masks (mask condition) and sunglasses (sunglasses condition). In total, there were 252 facial stimuli: 2 (sex) × 6 (individuals) × 7 (emotions) × 3 (uncovered vs. mask vs. sunglasses). All factors were within-subject factors.

## Procedure

The experiments were conducted in individually assigned cubicles, where the participants were briefly informed about the study and signed an informed consent form. During the experiment, participants were instructed to recognize facial expressions in a picture in which the facial stimuli were presented randomly across all factors (sex, emotion type, and condition). The stimuli were presented using E-Prime 3.0. The facial expression stimuli were presented on the left side of the monitor and a 3 × 3 table on the right side, with each emotional example entered as corresponding positions up to seven on a numeric keyboard. Participants were instructed to respond as fast and accurately as possible by pressing the numeric key that was aligned to the facial expression represented by the presented picture. An example of this task is shown in Fig 1.

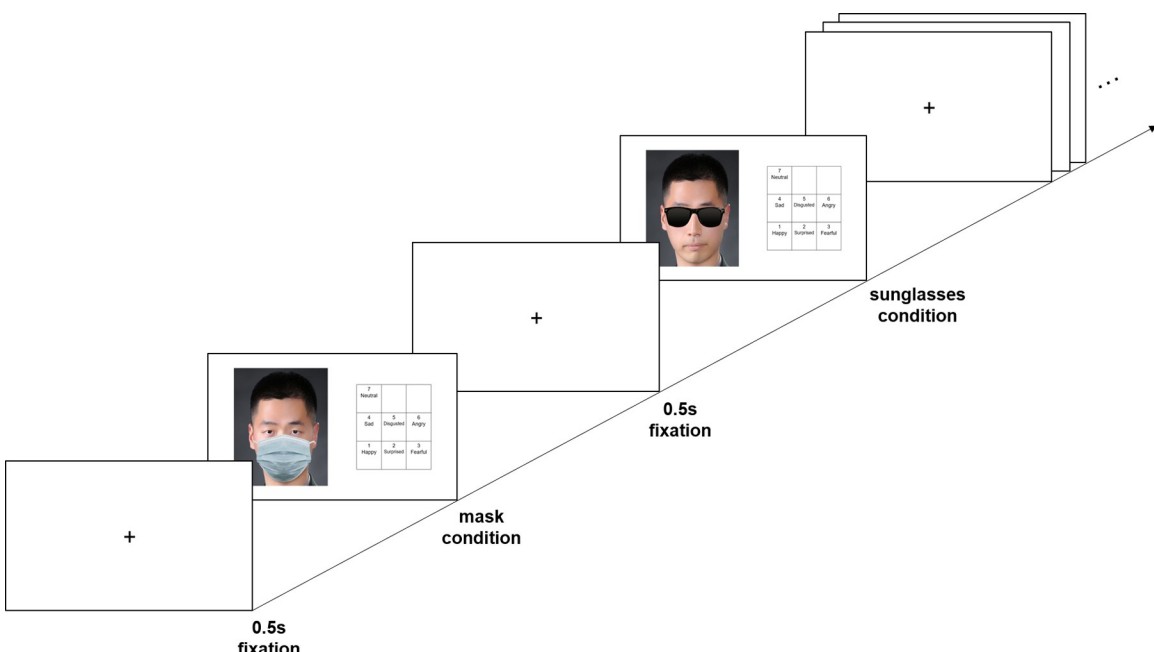

**Fig 1. An example of the facial expression recognition task.** Participants chose the emotional example that aligned to the facial expressions presented after 0.5-second fixation. All facial stimuli were presented randomly. Due to copyright, the presented face is an example photo, not KUFEC.

The participants completed six practice trials and then performed 252 test trials. After completion of the test trials, participants responded to a questionnaire regarding demographic information, were involved in a debriefing of the study, and received their participation rewards.

## Results

We tested the accuracy of the participants' baseline recognition rates by presenting the uncovered faces. Taking into consideration the chance rate of 14.2%, we can safely assume that the average correct answer rate of 68% for a fully visible face was well above the chance level ($\chi^2$ = 21312.458, $p < .001$, Cramer's V = 0.601, $p < .001$). We tested whether there were any sex or age effects on facial recognition accuracy. Sex differences in the overall correct answer rate were not statistically significant ($t$ -1.303, $p = .203$), nor was there a significant relationship between age recognition accuracy ($F = .032$, $p = .86$).

Before analyzing the main results, all data were checked for normality distribution. Normality was violated for some variables but we decided to use the repeated measures ANOVA because ANOVAs are generally considered robust to nonnormality with sample sizes being equal (within-subject design) [35]. A 3 (face condition) × 7 (emotion type) within-subjects ANOVA was conducted to test the main and interaction effects of the two variables: face regions that were covered and type of emotions. We reported the results of Greenhouse–Geisser correction whenever Mauchly's test of sphericity was significant. First, we predicted that the accuracy would be lower for recognizing faces wearing masks than for uncovered faces. The results showed that there was a significant main effect of face condition, $F(2, 76) = 147.34$, $p < .001$, $\eta_p^2 = .80$, with the highest recognition rate observed for the uncovered condition ($M = .68$, $SD = .01$), followed by the sunglasses ($M = .64$, $SD = .01$) and mask conditions ($M = .51$, $SD = .01$). Specifically, pairwise comparisons demonstrated that the accuracy under the mask condition was lower than both the uncovered condition, $t(38) = -15.98$, $p < .001$, and the sunglasses condition, $t(38) = -11.4$, $p < .001$. Not surprisingly, recognition accuracy under the sunglasses condition was lower than that under the uncovered condition, $t(38) = -4.74$, $p < .001$. Covering parts of the face, whether it is the eyes or mouth, hinders the performance of facial expression recognition. In particular, covering the mouth rather than the eyes made it more difficult to identify emotions.

There was also a significant main effect of emotion, using the Greenhouse-Geisser correction, $F(4.04, 153.44) = 211.96$, $p < .001$, $\eta_p^2 = .85$. Specifically, happiness and neutral emotion had higher recognition rates compared to all other emotions ($ts \geq 3.45$, $ps \leq .012$; $| ts | \geq 3.78$, $ps \leq .004$, respectively), whereas fear showed the lowest recognition rates ($| ts | \geq 12.28$, $ps < .001$). In summary, facial expression recognition accuracy was high in the order of happiness, neutral—surprise—sadness, disgust, anger—fear.

### Do different face regions (eye vs. mouth) make a difference in recognizing specific emotions?

In the previous analysis, participants' emotion recognition was significantly impaired when the mouth (and also a part of the nose) was covered (mask condition) than when the eyes were covered (sunglasses condition). Next, we tested whether the impaired facial recognition by mouth occlusion (vs. eye occlusion) would depend on specific emotional expressions in the face. Based on prior research, we expected that emotions such as happiness and disgust, the recognition of which were more influenced by the mouth (vs. the eyes) would be more impaired by wearing a mask than sunglasses; in contrast, we expected that for emotions such

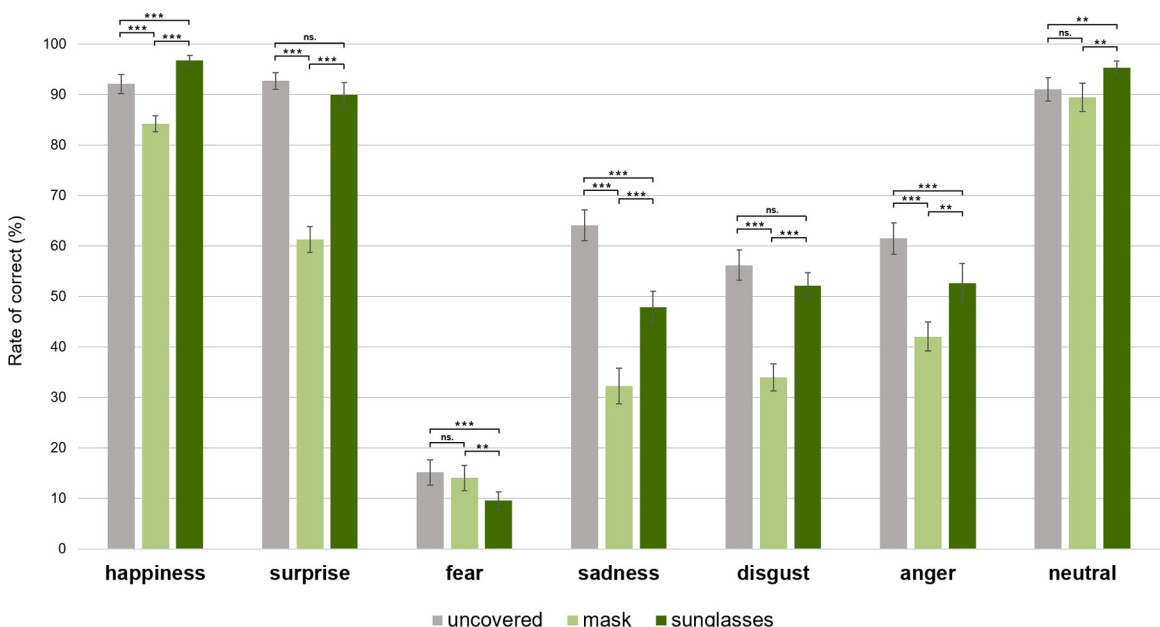

**Fig 2. Facial recognition accuracy across conditions and emotions.** Mean percentage of correct response across conditions and emotions. Error bars show the within-subjects standard error. Asterisks indicate statistical differences between conditions on the basis of pairwise comparisons (paired t-test): $^{**}p < 0.05$; $^{***}p < 0.001$; ns, not significant.

as fear, for which the eye has a higher impact on emotion recognition, would be less recognized by wearing sunglasses compared to a mask.

As shown in Fig 2, the interaction between face conditions and emotions was significant, $F$ (7.29, 277.04) = 15.22, $p < .001$, $\eta_p^2 = .29$. To interpret the interaction between the face conditions and the emotion type, we conducted pairwise comparisons by face condition in each emotion and examined whether the differences in accuracy rates between the mask, sunglasses and uncovered conditions would depend on the specific emotions. For instance, emotions such as happiness and disgust, which require cues primarily from the mouth, will show particularly lower accuracy under mask conditions than under sunglasses conditions. The results are as follows:

**Happiness.** The recognition accuracy of happiness under the mask condition ($M = .84$, $SE = .02$) was lower than that of both the sunglasses ($M = .97$, $SE = .01$), $t(38) = -8.8$, $p < .001$, and the uncovered conditions ($M = .92$, $SE = .02$), $t(38) = -3.58$, $p = .001$. The recognition accuracy of the faces with sunglasses was higher than that of the uncovered faces, $t(38) = 2.76$, $p = .009$. This result was consistent with prior research that documented that the mouth is especially informative in the identification of happiness. However, it is noteworthy that participant accuracy in the sunglasses condition, which covered the eyes and left the mouth visible, was higher than that in the uncovered face condition. A possible reason for why people recognized happy faces with sunglasses better, and not worse, than uncovered happy faces may be because this allowed participants to concentrate only on the mouth without being distracted by the eyes. Previous studies have demonstrated that facial recognition performance improves by covering less important parts of the face [25]. When including both the mask and the sunglasses conditions, we were able to show that covering relatively less important areas in a face can actually increase recognition performance of certain emotions. Specifically, the present findings showed that people can perceive happiness by focusing only on the mouth, and information from the eyes interferes, rather than facilitates, emotion recognition.

**Disgust.**   For disgust, participants were less accurate when faces were covered by a mask ($M$ = .34, $SE$ = .03) than when they were uncovered ($M$ = .56, $SE$ = .03), $t(38)$ = -6.31, $p$ < .001, and when sunglasses were worn ($M$ = .52, $SE$ = .03), $t(38)$ = -5.97, $p$ < .001. Recognition accuracy for uncovered faces and sunglasses did not differ, $t(38)$ = -1.42, $p$ = .164. Considering that there was no significant difference between the sunglasses and uncovered conditions in disgust, disgust seems to be recognized from the mouth only, and the eyes do not play a major role.

**Sadness.**   Sad faces with masks ($M$ = .32, $SE$ = .04) showed lower detection accuracy than uncovered faces ($M$ = .64, $SE$ = .03), $t(38)$ = -9.13, $p$ < .001, and faces with sunglasses ($M$ = .48, $SE$ = .03), $t(38)$ = -4.15, $p$ < .001. Recognition accuracy under the sunglasses condition was also lower than that under the uncovered condition, $t(38)$ = -6.34, $p$ < .001. That is, it was particularly difficult to recognize sadness when the mouth was covered compared to when the eyes were covered.

**Anger.**   As for anger, faces covered by masks showed lower accuracy ($M$ = .42, $SE$ = .03) than uncovered faces ($M$ = .62, $SE$ = .03), $t(38)$ = -5.58, $p$ < .001, and those covered by sunglasses ($M$ = .53, $SE$ = .04), $t(38)$ = -2.68, $p$ = .011. Participants recognized angry faces with sunglasses less accurately than uncovered faces, $t(38)$ = -2.75, $p$ = .009. Thus, as with sadness, facial expression recognition of anger showed a significant decrease in accuracy in both the sunglasses and mask conditions, but especially in the mask condition. In other words, when the eyes, but not the mouth, were covered, people could still make emotional inferences of sad or angry faces from the mouth to some degree. Furthermore, the lower accuracy in the sunglasses condition compared to the uncovered condition further corroborates that people detect sadness and anger from the eyes as well.

**Surprise.**   As for the emotion of surprise, participants recognized the emotions in the masked condition less accurately ($M$ = .61, $SE$ = .03) than in the uncovered condition ($M$ = .93, $SE$ = .02), $t(38)$ = -12.05, $p$ < .001, and the sunglasses condition ($M$ = .9, $SE$ = .02), $t(38)$ = -10.35, $p$ < .001. The recognition accuracy of uncovered faces and faces with sunglasses did not differ, $t(38)$ = 1.36, $p$ = .181. That is, people were able to recognize surprise by focusing on the mouth and did not gain much from focusing on the eyes.

**Fear.**   Interestingly, recognition of fear did not seem to be affected by the covering of the mouth. The recognition accuracy of fear under the mask condition ($M$ = .14, $SE$ = .03) was similar to the uncovered condition ($M$ = .15, $SE$ = .03), $t(38)$ = -.53, $p$ = .602. However, accuracy under the sunglasses condition ($M$ = .1, $SE$ = .02) was lower than that under the other two conditions, $| ts | \geq 2.14$, $ps \leq .039$. This suggests that covering the eyes, but not the mouth, decreased recognition. Consistent with previous results, people tend to recognize fearful faces from the eyes (upper facial area) [9, 12].

**Neutral.**   Finally, neutral faces with sunglasses showed higher recognition rates ($M$ = .95, $SE$ = 0.01) compared to uncovered faces ($M$ = .91, $SE$ = .02), $t(38)$ = 2.4, $p$ = .021, and faces covered by masks ($M$ = .90, $SE$ = .03), $t(38)$ = 2.61, $p$ = .013. The recognition accuracy of neutral expressions under the mask and uncovered conditions did not differ, $t(38)$ = .937, $p$ = .355.

## How were the emotions misinterpreted?

We further analyzed how participants incorrectly recognized the emotions of a face covered by a mask. This will allow us to better understand how the lack of cues from masked faces is dealt with and provides real-world implications for individuals living during a pandemic. The percentages of correct and incorrect answers to all emotions are presented in the confusion matrix of emotions (Fig 3).

The correct answer rate of sadness in an uncovered face was 64.1%, but the correct answer rate of the sad face wearing a mask fell dramatically to 32.3%, of which 20.7% of participants

**Confusion matrix of emotions**

| | | presented emotion | | | | | | |
|---|---|---|---|---|---|---|---|---|
| | | happiness | surprise | fear | sadness | disgust | anger | neutral | |
| recognized emotion | happiness | 92.1% | 0.9% | 0.2% | 3.2% | 0.4% | 0.6% | 3.8% | uncovered |
| | surprise | 3.6% | 92.7% | 65.0% | 0.4% | 5.1% | 3.4% | 0.2% | |
| | fear | 0.2% | 4.3% | 15.2% | 4.9% | 5.3% | 2.6% | 0.6% | |
| | sadness | 1.1% | 0.2% | 2.1% | 64.1% | 16.5% | 0.4% | 2.0% | |
| | disgust | 0.4% | 0.6% | 12.8% | 18.4% | 56.2% | 19.0% | 1.6% | |
| | anger | 1.3% | 0.2% | 3.8% | 3.4% | 15.4% | 61.5% | 1.5% | |
| | neutral | 1.3% | 1.1% | 0.9% | 5.6% | 1.1% | 12.4% | 91.0% | |
| | happiness | 84.2% | 4.5% | 1.9% | 7.9% | 7.9% | 1.7% | 4.7% | mask |
| | surprise | 4.3% | 61.3% | 40.8% | 1.9% | 1.5% | 10.9% | 1.5% | |
| | fear | 1.1% | 4.1% | 14.1% | 7.5% | 4.9% | 5.3% | 0.6% | |
| | sadness | 1.3% | 0.4% | 6.2% | 32.3% | 24.8% | 0.9% | 1.1% | |
| | disgust | 0.2% | 0.2% | 8.8% | 20.7% | 34.0% | 7.5% | 0.2% | |
| | anger | 0.9% | 0.6% | 15.4% | 10.3% | 23.7% | 42.1% | 2.4% | |
| | neutral | 8.1% | 28.8% | 12.8% | 19.4% | 3.2% | 31.6% | 89.5% | |
| | happiness | 96.8% | 1.1% | 0.6% | 2.8% | 0.2% | 1.7% | 1.5% | sunglasses |
| | surprise | 0.6% | 90.0% | 65.8% | 3.2% | 11.5% | 3.2% | 0.9% | |
| | fear | 0.0% | 1.9% | 9.6% | 3.2% | 2.6% | 1.7% | 0.2% | |
| | sadness | 0.6% | 0.0% | 2.4% | 47.9% | 8.8% | 1.3% | 0.6% | |
| | disgust | 0.2% | 0.6% | 12.6% | 23.9% | 52.1% | 25.2% | 0.9% | |
| | anger | 0.6% | 0.6% | 6.8% | 9.8% | 23.9% | 52.6% | 0.6% | |
| | neutral | 1.1% | 5.8% | 2.1% | 9.2% | 0.9% | 14.3% | 95.3% | |

**Fig 3. Percentage of participants responding to each emotion.** Rows correspond to the recognized emotion (participants' responses) and columns correspond to the presented (correct) emotion.

incorrectly recognized as disgust. As for the emotion of disgust, the correct answer rate for the uncovered faces was 56.2%, but it dropped substantially to 34% when the face was covered with a mask. In this case, 24.8% of participants mistakenly recognized it as sadness and 23.7% as anger. This suggests that without the facial configuration of the mouth, sadness can easily be confused with disgust. Consistent with the present study, Carbon [9] also demonstrated that sadness, disgust, and anger were confused with each other in masked faces. Additionally, sadness was detected with 64.1% accuracy under uncovered conditions; however, this decreased to 47.9% in the sunglasses condition, where 23.9% misrecognized sadness as disgust. Furthermore, in the case of anger, the 61.5% accuracy rate in the uncovered condition decreased to 52.6% under the sunglasses condition, and 25.2% misrecognized it as disgust and 14.3% as neutral. Therefore, sadness, disgust, and anger were likely to be confused with each other under sunglasses conditions as well.

As for surprise, the recognition rate was 61.3% in the mask condition, and 28.8% of participants misrecognized it as a neutral emotion. However, for the neutral faces with masks, the correct answer rate was 89.5%, with very few people mistaking these for other emotions. That is, wearing a mask increased the possibility of misrecognizing surprise as a neutral expression, but the reverse was not the case.

The other emotions discussed earlier (i.e., sadness, disgust, and anger) were likely to be confused with each other, but interestingly, the confusion between surprise and neutral emotions was asymmetrical. Interestingly, surprise was not confused with a neutral emotion in the sunglasses condition, as indicated by its high recognition rate of 90%; the misrecognition of surprise as a neutral emotion was only observed when the mouth was covered.

Another point to note in our findings is the misinterpretation of fear as surprise. Under the sunglasses condition, 66% of the participants incorrectly recognized fear as surprise. Moreover, 65% of the participants recognized fear as surprise when a face was uncovered, and these were often confused with one another [12, 36, 37]. In a study that used the KUFEC, participants were not able to make a clear distinction between surprise and fear [34].

## Does sex matter?

In this experiment, picture stimuli consisted of six male and six female models. We tested whether there was a difference in the recognition accuracy of facial expressions depending on the sex of the target face. We conducted a repeated-measures ANOVA with the sex of the face as a within-subject factor and the sex of the participant as a between-subject factor. The results showed that the main effect of the sex of the stimuli was statistically significant, $F(1, 37) = 5.87$, $p = .02$, $\eta_p^2 = .14$, indicating that the accuracy was higher when female faces (vs. male faces) were presented, $t(37) = 2.42$, $p = .02$. However, the main effect of the sex of the participants and the interaction between the sex of the participants and the sex of the face stimuli were not statistically significant, $F(1,37) = 1.74$, $p = .20$; $F(1, 37) = 1.42$, $p = .24$; respectively. Thus, participants were more accurate in recognizing the emotions of female faces, regardless of their sex.

## Discussion

In this experiment, we set out to test whether facial occlusion in real-world settings such as occlusion with facial masks and sunglasses impairs recognition of emotions in a face, with a particular focus on the effects of mask wearing, which is still mandatory in many parts of the world due to the COVID-19 pandemic. We did this by comparing the recognition of six basic emotions under masked conditions compared to sunglasses or uncovered conditions. We found that the recognition rates for faces with masks and sunglasses were lower than for the uncovered faces. Wearing a mask particularly harmed the recognition of emotional expressions in the face, as indicated by a greater decrease in recognition under the mask condition compared to under the sunglasses condition. Specifically, happiness, surprise, sadness, disgust, and anger showed the lowest recognition accuracy when the mouth was covered (i.e., mask condition), suggesting the important role played by the mouth in these emotions. Fear, on the other hand, showed the lowest recognition accuracy when the eyes were covered (i.e., the sunglasses condition). As for the neutral face, there was no significant difference between the uncovered and masked faces. Taken together, our findings suggest that facial masks cause the most striking decline in recognition of the majority of basic emotions among Koreans, which is consistent with prior research on Western participants. In addition to replicating prior studies, the present study has several noteworthy findings.

By simultaneously comparing faces with different parts covered, the present study found that covering certain parts of the facial areas *increases*, not decreases, recognition of some emotions. For instance, when the eyes were covered, happiness was recognized better than when faces were uncovered, suggesting that covering the upper facial area, which is not critical for perceiving happiness, could actually facilitate emotion recognition. Moreover, with the addition of the sunglasses condition in the present study, the critical role of the mouth in emotion recognition was confirmed in emotions such as surprise and disgust, as similar accuracy rates were observed for faces with sunglasses and uncovered faces. On the other hand, emotions such as sadness and anger in faces with sunglasses were not as recognizable as in uncovered faces, indicating that people attain facial expression information for these emotions from the eyes in addition to the mouth.

In addition to examining the overall recognition rate of different emotions when different parts of the face are covered, it is worthwhile to have a closer look at how participants misinterpreted the emotional expressions of these faces. Specifically, negative emotions, including sadness, disgust, and anger, were often confused with one another. In particular, when the mouth was covered (i.e., mask condition), sadness and disgust were often misidentified as one another. That is, without any cues from the mouth, people have trouble distinguishing these emotions. These findings offer practical advice for people who communicate while wearing a face mask. For instance, individuals who are sad may benefit from knowing that people may misrecognize their sadness as disgust. Likewise, it would be helpful for people to know that what seems like a disgusted face may actually be a sad face.

Finally, to our knowledge, this is the first study to test the effect of mask wearing on emotion recognition with a direct comparison to the effects of sunglasses among East Asians. It is widely known that East Asians focus on the eyes while Westerners focus on the mouth when recognizing facial expressions. Thus, we expected a decrease in recognition accuracy in the sunglasses condition, as the eyes are especially important in recognizing facial expressions for East Asians. The results showed that there was indeed a decrease in the recognition of faces with sunglasses. However, the largest decrease in recognition was observed for the masked faces, more than the faces with sunglasses, which is consistent with Noyes et al. [17], suggesting that the mouth is the most important source of information for most basic emotions for Koreans as well. The overall facial expression recognition accuracy was similar to that of Westerners [9, 17], but some emotions showed different patterns. For instance, in our findings, the mask had destructive effects on sad faces, but not on fearful faces, which was consistent with Germans [9] but not with British people [17]. Specifically, comparing faces covered with masks and faces covered with sunglasses, our participants recognized fearful faces more accurately in mask conditions and sad faces in sunglasses conditions, while British participants showed the opposite patterns. In addition, our participants had difficulty classifying surprise and neutral states from the faces who wore shades compared to uncovered faces, but there was no difference in British participants. This suggests that East Asians retrieve information about emotions from eyes more than Westerners when reading facial expressions of surprise. However, there was also a finding that Westerners, rather than East Asians, read emotions from eyes. For instance, our participants showed higher accuracy for angry faces in sunglasses than mask conditions, but UK participants showed no significant difference between these conditions, suggesting that the mouth area was more informative for Koreans, but both mouth and eye areas were informative for British people. However, since these participants were not collected at the same time with the same procedures, the current study cannot directly compare with British people, making it difficult to determine the influence of culture. Therefore, future research that directly compares the differences between East Asian and Western participants is needed.

Finally, participant gender did not have any effect on recognition rate nor did they interact with the target's gender. However, the target stimuli's gender mattered. Overall, participants recognized female faces better than male faces. More research is needed to replicate this effect.

### Limitations and future research

Some limitations of our study should be considered in future research. First, because the participants in our study were limited to undergraduate students, possible age effects were not tested. Some studies suggest that people have more difficulty recognizing older faces than middle-aged or younger faces [9, 38], and that middle-aged participants identify facial expressions better than children or older adults [39]. Future research should therefore examine the effects of the age of both the raters and the target faces with realistic facial occlusions. Second, the

present study used still photos of facial expressions, which is far from the real-world communication of facial expressions. Recently, researchers measured the accuracy of emotional recognition using video stimulation, which adds a static background to the dynamic facial expression set [40]. However, since this study included only two emotions, happiness and sadness, it is difficult to grasp the effect of on recognition of various emotions in facial expression. Moreover, the face is not the only channel through which emotions are recognized. Other information such as context [41], body language [42, 43], and voice [44] are also used to express or read emotions. Thus, future studies using video stimuli that include various emotions and involve other sources of information would render a more ecologically valid effect of facial occlusion on emotion recognition.

Wearing a facial mask will not go out of fashion anytime soon, as COVID-19 is unlikely to cease and continues to stay with us as an endemic influenza [45]. Given this situation, the present line of research could be developed so that we can find ways in which people can adapt to restricted communication due to facial masks. An intervention that targets specific groups that are particularly vulnerable to the current situation is a good example. For instance, service employees who are required to recognize customers' facial expressions while masks are worn are especially at a disadvantage. Given that tourism and hospitality staff showed an increase in both accuracy and speed of facial expression recognition [46], a similar training would enhance emotion recognition for masked faces as well. Another group that requires special attention is children. In recent studies, researchers have found that children's emotion recognition is also affected by facial masks [15, 16, 47]. Given that childhood is an important developmental stage for the socialization of emotion [48, 49] and a potential influence of mask wearing on the development of necessary social interaction skills [18], it would be important not only to understand the effect of facial occlusion among children but also to develop strategies to help young children learn to read emotions from others.

In conclusion, our results suggest that the mouth is more important than the eyes in facial expression recognition. As mask wearing is becoming increasingly important, future studies should investigate effective ways, such as interventions, to minimize miscommunication from incorrectly reading emotions from a face.

## Acknowledgments

We would like to thank Dr. June-Seek Choi for letting us use the KUFEC stimuli set for this study.

## Author Contributions

**Conceptualization:** Garam Kim.

**Formal analysis:** Garam Kim, Seok-Sung Hong.

**Funding acquisition:** Eunsoo Choi.

**Investigation:** Garam Kim, So Hyun Seong.

**Methodology:** Garam Kim, Seok-Sung Hong, Eunsoo Choi.

**Project administration:** Eunsoo Choi.

**Resources:** Seok-Sung Hong, Eunsoo Choi.

**Visualization:** So Hyun Seong.

**Writing – original draft:** Garam Kim, So Hyun Seong.

**Writing – review & editing:** Garam Kim, So Hyun Seong, Eunsoo Choi.

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
