## [Decision Letter · Decision Letter 0]

22 Sep 2021

PONE-D-21-25701Is wearing a mask masking one's emotion?PLOS ONE

Dear Dr. Choi

Thank you for submitting your manuscript to PLOS ONE. After careful consideration, we feel that it has merit but does not fully meet PLOS ONE’s publication criteria as it currently stands. Therefore, we invite you to submit a revised version of the manuscript that addresses the points raised during the review process. We have now feedback from Reviewer ! who is an expert in the field, and (since we have some difficulties in securing reviewers due to summer time) I had a close look at your submission. Both of us find your work interesting. Yet we both require a major revision of your manuscript. Reviewer 1 provided you with a thorough review and a number of major and minor concerns and suggestions (please find below). My main concerns are:1) You have update your Introduction and Discussion taking into account latest studies in the field. Our analysis Pavlova and Sokolov READING COVERED FACES. Cerebral Cortex 2021 may be of substantial help; 2) I agree with Reviewer 1 that the title should be modified to better reflect the specificity and novelty of your study. Please carefully consider what is new in your work as compare with previous research; 3) IMPORTANT; did you check the data sets for normality of distribution and how? Please make a cklear statement on it in the Method Section. Parametric statistics can be used only if the data sets are normally distributed, otherwise, please use non-parametric statistics; 4) I appreciate that you analysed the effect of a poser gender. But did you analyse the effect of observers' gender on performance? PLease address all issues on the point-by-point basis in your rebuttal letter to me. 

We look forward to receiving your revised manuscript.

Kind regards,

Marina A. Pavlova, PhD

Academic Editor

PLOS ONE

“E.C received the grant from School of Psychology, Korea University (K2110491).”            

“NO authors have competing interests”

 This information should be included in your cover letter; we will change the online submission form on your behalf

Reviewers' comments:

Reviewer's Responses to Questions

**Comments to the Author**

1. Is the manuscript technically sound, and do the data support the conclusions?

Reviewer #1: Yes

2. Has the statistical analysis been performed appropriately and rigorously? 

Reviewer #1: Yes

3. Have the authors made all data underlying the findings in their manuscript fully available?

Reviewer #1: Yes

4. Is the manuscript presented in an intelligible fashion and written in standard English?

Reviewer #1: Yes

5. Review Comments to the Author

Reviewer #1: The general topic of this paper is still timely as the pandemic is seemingly never ending, so research about the impact of wearing masks is relevant, and emotional reading in particular is one cognitive function which is obviously strongly impoverished when a main part of the face is covered.

In particularly liked that the authors employed Korean images and added a further, ecological relevant variable by using sunglasses as a kind of control condition to face masks to test effects of mouth vs. eyes-covering. Additionally I appreciate that they offered high quality analyses which are nicely depicted. The paper is clear, only the introduction seems to miss some points.

I will comment on this in detail in my report to the authors below.

Major points

1. The authors claim that the Ruba & Pollack study was “particularly informative”; I would like to disagree, and the authors sum this up competently later on: That study was a (very) limited one; additional to the critical points the authors already raised, I would like to add that the material and experimental design was highly problematic as we cannot assign the variance to the specific face or the specific intervention (type of covering) as different faces were used across conditions—in the end, the authors devote a very long section on that paper but the validity of it is currently debated. Please consider new studies which addressed these critical points, e.g. Carbon & Serrano (2021).

2. G_Power-analyses: this makes sense to conduct such an analysis before conducting the study, but why then were 39 instead of 28 persons tested. Was the test power analysis in the end just done for the sake of having done it? Please verify this.

3. Title: this is typically NOT a major issue, but here the title is so underspecified that I would not recommend it (and I do this to protect the authors, therefore raising it to “major” as I believe that more informative papers will help to be read by others—and I truly believe that this paper deserves many readings by interesting readers!).

Minor points

1. Please reduce the “precision” with values which are imprecisely measured, e.g. age.

2. Please add units to values (e.g., again, age)

3. Greek letters should not be italicized

4. It is great that you have offered the data on OSF! Please consider (later on) to provide the analyses, too

5. Figure 1 is very illustrative, thank you! I would extend the arrow and put a “…” at the end, juts to make clear it goes on and on and that a trial is NOT a couple of two pictures but that this diagram just shows a sequence of two typical trials (the quality of the scale could be increased in the final version—it is hard to read currently).

6. Please insert spaces between values and units

7. Please add effect sizes, also to the Chi2-tests

8. Fig.2 is very nice, but please add the method how you analysed the pairwise comparisons; was it alpha-corrected? How were the error bars calculated? What do they represent? Just make everything clear, please!

9. “subject”: I would (just a recommendation, no rule or obligation—personal taste) not speak of “subjects” but persons / participants / etc.-

10. Discussion: About video material: please check the literature as there are already papers on video-related mask research

11. Baron-Cohen’s study [see ref.1] was additionally tested by Schmidtmann and colleagues with non-clinical participants meanwhile. Please check.

12. Discussion: children: see the latest paper on children-related samples

6. PLOS authors have the option to publish the peer review history of their article (what does this mean?). If published, this will include your full peer review and any attached files.

Reviewer #1: No

---

## [Author Response · Author response to Decision Letter 0]

9 Nov 2021

Dear Managing Editor,

PLOS ONE Reviews

Thank you for inviting us to submit a revised draft of our manuscript entitled, “Does masking one's facial area masks emotions? The impact of face masks and sunglasses on emotion recognition in South Koreans (revision)” to PLOS ONE. We express our cordial gratitude for the valuable and constructive review of our paper. The reviewers’ comments have helped us to further strengthen the overall quality of the paper. We have incorporated changes that reflect the detailed suggestions you have graciously provided. We also hope that our edits and the responses we provide below satisfactorily address all the issues and concerns you and the reviewer have noted. The specific responses to the reviewer’s comments are listed below.

The Editor’s comments:

Comment 1: You have update your Introduction and Discussion taking into account latest studies in the field.

Response: 

Thank you for the suggestion and recommending the recent papers on this topic. We have updated the introduction in the following manner:

We cited the paper by Pavlova and Sokolov (2021) and introduced the most recent relevant studies that were reviewed in the paper. Specifically, we incorporated the findings from Carbon & Serrano (2021) and Noyes et al., (2021) which were the most up-to-date studies that tested the effects of masked faces on emotion recognition. In addition, following reviewer 1’s suggestion, we have significantly reduced the part on the findings of Ruba & Pollack’s (2020) study. Furthermore, we updated the findings that gender differences exist in facial expression recognition of covered faces with the latest study synthesizing face masks.

In the discussion section, we provided a more comprehensive discussion by comparing our findings with Noyes et al. (2021) as well as Carbon (2020). In addition, we updated the latest studies in regard to reading emotions among children.

Comment 2: the title should be modified to better reflect the specificity and novelty of your study.

Response: As per your suggestion, we have changed the title to “Does masking one's facial area masks emotions? The impact of face masks and sunglasses on emotion recognition in South Koreans”. The manuscript title has been updated and now it is more meaningful.

Comment 3: did you check the data sets for normality of distribution and how? Please make a clear statement on it in the Method Section.

Response: Thank you for your suggestion. We tested normality of distribution and found that some variables were not normally distributed. However, we decided to continue using ANOVA as of the robustness of the statistic against violations of normality with equal sample sizes (Glass et al., 1972). The ANOVA is reasonably robust to violations of homogeneity of variance when group sizes are equal, and minor violations of the normal distribution can also still produce results that are similar to those when the data are normally distributed (Schmider et al. 2010). We have confirmed that many studies use the repeated measures ANOVA based on F test's robust to non-normality, even if the data are nonnormal (Ackermann et al., 2019; Adriaens et al., 2018; Everman et al., 2018; Ghazali et al., 2018; Goheen et al., 2013; Savage et al., 2020; Zhang et al., 2018). We also described that the violation of normality assumption and the reason we still use within-subject ANOVA in the Results section (p. 9, lines 184-187). Nevertheless, if you still think to need a nonnormality test, we will proceed with a new analysis. 

“Before analyzing the main results, all data were checked for normality distribution. Normality was violated for some variables but we decided to use the repeated measures ANOVA because ANOVAs are generally considered robust to nonnormality with sample sizes being equal (within-subject design) [34].”

Additionally, we added a sentence related to how to present results according to the tests of sphericity (p. 9, lines 189-190).

“We reported the results of Greenhouse–Geisser correction whenever Mauchly’s test of sphericity was significant.”

References

Glass G V, Peckham PD, & Sanders JR. Consequences of failure to meet assumptions underlying the fixed effects analyses of variance and covariance. Review of educational research. 1972;42(3): 237-288.

Schmider E, Ziegler M, Danay E, Beyer L, Bühner M. Is it really robust?. Methodology. 2010 Sep 8.

Ackermann C, Beggiato M, Bluhm LF, Löw A, Krems JF. Deceleration parameters and their applicability as informal communication signal between pedestrians and automated vehicles. Transportation research part F: traffic psychology and behaviour. 2019;62: 757-768.

Adriaens K, Van Gucht D, Baeyens F. IQOSTM vs. e-cigarette vs. tobacco cigarette: a direct comparison of short-term effects after overnight-abstinence. International journal of environmental research and public health. 2018;15(12): 2902.

Everman ER, Delzeit JL, Hunter FK, Gleason JM,Morgan TJ. Costs of cold acclimation on survival and reproductive behavior in Drosophila melanogaster. PLOS ONE. 2018;13(5): e0197822.

Ghazali AS, Ham J, Barakova E, Markopoulos P. The influence of social cues in persuasive social robots on psychological reactance and compliance. Computers in Human Behavior. 2018;87: 58-65.

Goheen JR, Palmer TM, Charles GK, Helgen KM, Kinyua SN, Maclean JE, ... Pringle RM. Piecewise disassembly of a large-herbivore community across a rainfall gradient: the UHURU experiment. PLOS ONE. 2013; 8(2): e55192.

Savage MJ, James R, Magistro D, Donaldson J, Healy LC, Nevill M, Hennis PJ. Mental health and movement behaviour during the COVID-19 pandemic in UK university students: Prospective cohort study. Mental Health and Physical Activity. 2020;19: 100357.

Zhang Z, Zhang B, Cao C, Chen W. The effects of using an active workstation on executive function in Chinese college students. PLOS ONE. 2018;13(6): e0197740.

Comment 4: Did you analyze the effect of observers' gender on performance?

Response: Yes. We described this at the beginning of the Results section. However to express it more effectively, we have revised the text to “However, the main effect of the sex of the participants and the interaction between the sex of the participants and the sex of the face stimuli were not statistically significant, F(1,37) = 1.74, p = .20; F(1, 37) = 1.42, p = .24; respectively.” (p. 17, lines 338-341).

The Reviewer's comments:

Comment 1: The authors claim that the Ruba & Pollack study was “particularly informative”; I would like to disagree, and the authors sum this up competently later on: That study was a (very) limited one; additional to the critical points the authors already raised, I would like to add that the material and experimental design was highly problematic as we cannot assign the variance to the specific face or the specific intervention (type of covering) as different faces were used across conditions—in the end, the authors devote a very long section on that paper but the validity of it is currently debated. Please consider new studies which addressed these critical points, e.g. Carbon & Serrano (2021).

Response: Thank you for pointing this out. The editor also expressed the same concern. Based on your and the editor’s recommendation, we have updated the introduction in the following manner:

We cited the paper by Pavlova and Sokolov (2021) and introduced the most recent relevant studies that were reviewed in the paper. Specifically, we incorporated the findings from Carbon & Serrano (2021) and Noyes et al., (2021) which were the most up-to-date studies that tested the effects of masked faces on emotion recognition. In addition, following reviewer 1’s suggestion, we have significantly reduced the part on the findings of Ruba & Pollack’s (2020) study. Furthermore, we updated the findings that gender differences exist in facial expression recognition of covered faces with the latest study synthesizing face masks.

In the discussion section, we provided a more comprehensive discussion by comparing our findings with Noyes et al. (2021) as well as Carbon (2020). In addition, we updated the latest studies in regard to reading emotions among children.

Comment 2: G_Power -analyses: this makes sense to conduct such an analysis before conducting the study, but why then were 39 instead of 28 persons tested. Was the test power analysis in the end just done for the sake of having done it? Please verify this.

Response: First, we calculated the number of participants required in order to meet the assumptions of independence and normality in our experiment using G-power. After that, we recruited more participants than we needed in anticipation of dropouts during the experiment or malfunctioning experimental computers or programs. In addition, more participants were recruited for more stable results because there may be participants who did not meet the pre-set exclusion criteria (less than a 50% correct facial expression recognition answer rate when presented with a fully visible face without a face mask or sunglasses were excluded). A total of 40 participants were recruited, but all 39 were used in the analysis because there was no reason to exclude them from the data except one. (p. 6, lines 130-134).

“However, we sampled more participants than the required number for potential participants who would get excluded if computers malfunction or those who did not meet the pre-set exclusion criteria (less than a 50% correct facial expression recognition answer rate when presented with a fully visible face without a face mask or sunglasses were excluded). This pre-determined criterion was based on that of Carbon’s method [9].”

Comment 3: Title : this is typically NOT a major issue, but here the title is so underspecified that I would not recommend.

Response: As per your suggestion, we have changed the title to “Does masking one's facial area masks emotions? The impact of face masks and sunglasses on emotion recognition in South Koreans”.

Comment 4: Please reduce the “precision” with values which are imprecisely measured, e.g. age.

Response: We did not understand this comment. Could you please explain what you meant by “reduce the precision”? Thank you.

Comment 5: Please add units to values (e.g., again, age).

Response: We have reflected this comment by adding units to age (p. 2, lines 23; p. 7, line 143).

“mean age = 24.23 years”

“Mage = 24.23 years, SDage = 4.68 years”

Comment 6: Greek letters should not be italicized.

Response: Thank you for your comments. We have revised the reporting of all partial eta-squared (ηp2) statistics throughout the paper. You can check them without italics.

Comment 7: It is great that you have offered the data on OSF! Please consider (later on) to provide the analyses, too

Response: Thank you for your suggestion. We will update with the syntax for the data analyses before the paper is published.

Comment 8: Figure 1 is very illustrative, thank you! I would extend the arrow and put a “…” at the end, juts to make clear it goes on and on and that a trial is NOT a couple of two pictures but that this diagram just shows a sequence of two typical trials (the quality of the scale could be increased in the final version—it is hard to read currently).

Response: We have now figure 1 with arrows and “…”. We think these changes now better to describe our trials. We hope that you agree.

Comment 9: Please insert spaces between values and units

Response: We checked it throughout the paper.

Comment 10: Please add effect sizes, also to the Chi2-tests.

Response: We calculated effect size of the χ2 using Cramer's V which is for categorical comparisons. Cramer's V coefficient and p-value were reported (p. 9, lines 180).

“(χ^2 = 21312.458, p < .001, Cramer’s V = 0.601, p < .001)”

Comment 11: Fig.2 is very nice, but please add the method how you analysed the pairwise comparisons; was it alpha-corrected? How were the error bars calculated? What do they represent? Just make everything clear, please!

Response: We computed a repeated-measures ANOVA and pairwise comparisons using SPSS 25 software, and constructed Fig 2 using the alpha and standard errors provided by the package. We agree with you and have clarified how we analyzed (under the Fig.2). We think this change now is better for readers to understand our analysis. We hope that you agree.

Comment 12: “subject”: I would (just a recommendation, no rule or obligation—personal taste) not speak of “subjects” but persons / participants / etc.

Response: We have reflected this comment by replacing the term “subject” throughout the paper with “participants”.

Comment 13: Discussion: About video material : please check the literature as there are already papers on video-related mask research.

Response: Thank you for the reminder. As you have mentioned about existing literature on video materials, we managed to find that a paper with regards to video materials with faces wearing masks and thus, included it in our Discussion section. However, since the paper has limitations in that, it contains only two emotions, happy and sad, we decided to maintain the argument that video research is necessary for the effect of masks on facial expressions recognition (p. 20, lines 418-422; 424-426).

“Recently, researchers measured the accuracy of emotional recognition using video stimulation, which adds a static background to the dynamic facial expression set [40]. However, since this study included only two emotions, happiness and sadness, and sad, it is difficult to grasp the effect of on recognition of various emotions in facial expression.”

“Thus, future studies using video stimuli that include various emotions and involve other sources of information would render a more ecologically valid effect of facial occlusion on emotion recognition.”

Comment 14: Baron-Cohen’s study [see ref.1] was additionally tested by Schmidtmann and colleagues with non-clinical participants meanwhile. Please check.

Response: We checked the study of Schmidtmann et al. (2020) and changed the references. Thank you for your suggestion.

Comment 15: Discussion: children : see the latest paper on children-related samples

Response: We checked the paper (Carbon & Serrano, 2021) that examined the effect of masks on facial expression recognition in children and added it in the corresponding part. (p. 21, lines 436-437)

“In recent studies, researchers have found that children’s emotion recognition is also affected by facial masks [15, 16, 47].”

Again, thank you for giving us the opportunity to strengthen our manuscript with your valuable comments and queries. We have worked hard to incorporate your feedback and hope that these revisions persuade you to accept our submission.

---

## [Decision Letter · Decision Letter 1]

21 Dec 2021

PONE-D-21-25701R1Does masking one's facial area masks emotions? The impact of face masks and sunglasses on emotion recognition in South KoreansPLOS ONE

Dear Dr. Choi:

Thank you for submitting your manuscript to PLOS ONE. After careful consideration, we feel that it has merit but does not fully meet PLOS ONE’s publication criteria as it currently stands. Therefore, we invite you to submit a revised version of the manuscript that addresses the points raised during the review process.REviewer 1 is almost sasatisfied with your revision. However, he/she still express some concerns that shoulde be carefuölly addressed (e.g., issue 5 of his/her review). I would also suggest you look attentively at READING COVERED FACES, in particular, at cultural differences in reading masked faces analyzed there. It would be nice information ypou can use for your manuscript. 

We look forward to receiving your revised manuscript.

Kind regards,

Marina A. Pavlova, PhD

Academic Editor

PLOS ONE

Journal Requirements:

Reviewers' comments:

Reviewer's Responses to Questions

**Comments to the Author**

1. If the authors have adequately addressed your comments raised in a previous round of review and you feel that this manuscript is now acceptable for publication, you may indicate that here to bypass the “Comments to the Author” section, enter your conflict of interest statement in the “Confidential to Editor” section, and submit your "Accept" recommendation.

Reviewer #1: All comments have been addressed

2. Is the manuscript technically sound, and do the data support the conclusions?

Reviewer #1: Partly

3. Has the statistical analysis been performed appropriately and rigorously? 

Reviewer #1: Yes

4. Have the authors made all data underlying the findings in their manuscript fully available?

Reviewer #1: Yes

5. Is the manuscript presented in an intelligible fashion and written in standard English?

Reviewer #1: Yes

6. Review Comments to the Author

Reviewer #1: Dear Authors, thanks for providing such a precise and clear rebuttal letter which greatly supports my job as a reviewer. I found your replies convincing and well founded.

I would only recommend to address some (very!) minor points.

(very) minor points

1. The effect of gender was not significant as you explained in your letter; please state 1-2 sentences whether this was expected or not and on which basis you argue pro or contra such effects.

2. About Comment 4 (from my review before): please use only the precision the base of measurement provides, e.g. age is typically asked regarding years, but not more precise. So any outcome of mean age should not be more precise than about 1/10 of the original precision. So for instance 24.2 instead of 24.23 years.

3. [Just a “Thank you!”: Thanks for providing your data and analyses on OSF—the scientific community will appreciate this very much!]

4. Please leave out [Internet] for references

5. IMPORTANT POINT: Fig.3: PLEASE DOUBLE CHECK THIS (just because it is highly unrealistic): was fear really mostly misinterpreted as SURPRISE? I have never seen such a massive systematic confusion of emotional states across different presentation conditions.

7. PLOS authors have the option to publish the peer review history of their article (what does this mean?). If published, this will include your full peer review and any attached files.

Reviewer #1: No

---

## [Author Response · Author response to Decision Letter 1]

5 Jan 2022

1. The effect of gender was not significant as you explained in your letter; please state 1-2 sentences whether this was expected or not and on which basis you argue pro or contra such effects.

Response: Although gender was not a major concern of our study, we expected two main effects of gender based on previous studies. First, we expected that women would show higher recognition rates than men. Second, we expected the participants to perform better when the target face is of opposite sex. However, the results showed that the main effect of participant sex and the interaction between participant gender and the gender of target were not significant. Unexpectedly, there was a main effect of stimuli gender was significant. Specifically, we found that participants perceived the emotions of female (vs. male) faces more accurately. As we know of no theoretical reason for this result, more research is needed to replicate such findings first.

The following paragraph is added in the revised manuscript. 

“Finally, participant gender did not have any effect on recognition rate nor did they interact with the target’s gender. However, target stimuli’s gender mattered. Overall, participants recognized female faces better than male faces. More research is needed to replicate this effect.” (page 20, lines 410-413)

2. About Comment 4 (from my review before): please use only the precision the base of measurement provides, e.g. age is typically asked regarding years, but not more precise. So any outcome of mean age should not be more precise than about 1/10 of the original precision. So for instance 24.2 instead of 24.23 years.

Response: Thank you for your kind explanation. As per your suggestion, we have revised the precision of age by rounding up to the first digit after the decimal point instead of the second digit.

3. [Just a “Thank you!”: Thanks for providing your data and analyses on OSF—the scientific community will appreciate this very much!]

Response: Thank you for the reminder. We updated the syntax for the data analyses as well as our figures. 

4. Please leave out [Internet] for references

Response: We removed [Internet] from the references.

5. IMPORTANT POINT: Fig.3: PLEASE DOUBLE CHECK THIS (just because it is highly unrealistic): was fear really mostly misinterpreted as SURPRISE? I have never seen such a massive systematic confusion of emotional states across different presentation conditions.

Response: In this experiment, participants tended to recognize fear as surprise rather than as fear itself. (We were also surprised by these patterns, and we checked the results again several times.) Not only did 41% of participants incorrectly recognize fear as surprise in a mask condition, 65% of participants misread fear as surprise when in uncovered face. 

It is true that surprise and fear are often confused (Ekman, 2003) but in our study, the misinterpretation rate was very high. This may suggest that there the facial expression of "fear” in the present facial stimuli set is problematic. In line with this argument, in a study conducted by Kim et al. (2011) where the same KUFEC database was used, participants were not able to make a significant distinction between surprise and fear. 

Again, thank you for giving us the opportunity to strengthen our manuscript with your valuable comments and queries. We have worked hard to incorporate your feedback and hope that these revisions persuade you to accept our submission.

---

## [Decision Letter · Decision Letter 2]

10 Jan 2022

PONE-D-21-25701R2Does masking one's facial area masks emotions? The impact of face masks and sunglasses on emotion recognition in South KoreansPLOS ONE

Dear Dr.Choi,

Thank you for submitting your manuscript to PLOS ONE. After careful consideration, we feel that it has merit but does not fully meet PLOS ONE’s publication criteria as it currently stands. Therefore, we invite you to submit a revised version of the manuscript that addresses the points raised during the review process. I suggest to shorten the title of your paper to 'Impact of face masks and sunglasses on emotion recognition in South Koreans. You also compßletely ignored my request for more attentive reading of the recent review READING COVERED FACES, and for adding more information on cultural differences on recognition of covered by masks emotions. I hope you can do this promptly.

We look forward to receiving your revised manuscript.

Kind regards,

Marina A. Pavlova, PhD

Academic Editor

PLOS ONE

Journal Requirements:

Reviewers' comments:

Reviewer's Responses to Questions

**Comments to the Author**

1. If the authors have adequately addressed your comments raised in a previous round of review and you feel that this manuscript is now acceptable for publication, you may indicate that here to bypass the “Comments to the Author” section, enter your conflict of interest statement in the “Confidential to Editor” section, and submit your "Accept" recommendation.

Reviewer #1: All comments have been addressed

2. Is the manuscript technically sound, and do the data support the conclusions?

Reviewer #1: Yes

3. Has the statistical analysis been performed appropriately and rigorously? 

Reviewer #1: Yes

4. Have the authors made all data underlying the findings in their manuscript fully available?

Reviewer #1: Yes

5. Is the manuscript presented in an intelligible fashion and written in standard English?

Reviewer #1: Yes

6. Review Comments to the Author

Reviewer #1: Thanks for your quick makeover; all points are now addressed (thanks also for clarifying the issue with SURPRISE vs. FEAR) and I wish the authors alle the best for their paper!

7. PLOS authors have the option to publish the peer review history of their article (what does this mean?). If published, this will include your full peer review and any attached files.

Reviewer #1: No

---

## [Author Response · Author response to Decision Letter 2]

19 Jan 2022

The Editor's comments:

Comment 1: I suggest to shorten the title of your paper to 'Impact of face masks and sunglasses on emotion recognition in South Koreans. 

Response: As per your suggestion, we have shortened the title to “Impact of face masks and sunglasses on emotion recognition in South Koreans.” We hope that the change will make it easier for readers to access our paper.

Comment 2: You also completely ignored my request for more attentive reading of the recent review READING COVERED FACES, and for adding more information on cultural differences on recognition of covered by masks emotions. I hope you can do this promptly.

Response: We apologize for carelessly omitting the response to your comment regarding cross-cultural implications of the present research. The current manuscript has added more information on cultural differences on recognition of covered faces by citing studies from your review paper. We greatly appreciate your helpful suggestions. 

See p. 5, lines 101-109

" Also, the effects of masks or sunglasses on reading other person's facial expressions may differ depending on the cultural context (for a review see [18]). As an example, consider the findings that a face covered with Islamic headdress such as niqāb impacts the recognition of emotions differently by cultural groups [13, 25], suggesting that a culturally attached meaning of headdress may play a role. As for East Asians, they are not only less accustomed to sunglasses than Westerners, but sunglasses are often considered rude in interpersonal relationships [26, 27]. Given such cultural background, thus, it is necessary to test whether the effects of masks and sunglasses on facial expression recognition that are documented with Western participants would also apply to East Asians.”

---

## [Editor Report · Decision Letter 3]

20 Jan 2022

Impact of face masks and sunglasses on emotion recognition in South Koreans

PONE-D-21-25701R3

Dear Dr. Choi:

We’re pleased to inform you that your manuscript has been judged scientifically suitable for publication and will be formally accepted for publication once it meets all outstanding technical requirements.

Kind regards,

Marina A. Pavlova, PhD

Academic Editor

PLOS ONE
---

## [Editor Report · Acceptance letter]

24 Jan 2022

PONE-D-21-25701R3 

Impact of face masks and sunglasses on emotion recognition in South Koreans 

Dear Dr. Choi:

I'm pleased to inform you that your manuscript has been deemed suitable for publication in PLOS ONE. Congratulations! Your manuscript is now with our production department. 

Kind regards, 

on behalf of

Prof. Marina A. Pavlova 

Academic Editor

PLOS ONE